# Transmission phase read-out of a large quantum dot in a nanowire interferometer

Francesco Borsoi [1], Kun Zuo[2], Sasa Gazibegovic[3], Roy L. M. Op het Veld[3], Erik P. A. M. Bakkers [3], Leo P. Kouwenhoven[1,4] & Sebastian Heedt [1,4✉]

Detecting the transmission phase of a quantum dot via interferometry can reveal the symmetry of the orbitals and details of electron transport. Crucially, interferometry will enable the read-out of topological qubits based on one-dimensional nanowires. However, measuring the transmission phase of a quantum dot in a nanowire has not yet been established. Here, we exploit recent breakthroughs in the growth of one-dimensional networks and demonstrate interferometric read-out in a nanowire-based architecture. In our two-path interferometer, we define a quantum dot in one branch and use the other path as a reference arm. We observe Fano resonances stemming from the interference between electrons that travel through the reference arm and undergo resonant tunnelling in the quantum dot. Between consecutive Fano peaks, the transmission phase exhibits phase lapses that are affected by the presence of multiple trajectories in the interferometer. These results provide critical insights for the design of future topological qubits.

[1] QuTech and Kavli Institute of Nanoscience, Delft University of Technology, GA Delft 2600, The Netherlands. [2] RIKEN Center for Emergent Matter Science (CEMS), Wako, Saitama 351-0198, Japan. [3] Department of Applied Physics, Eindhoven University of Technology, MB Eindhoven 5600, The Netherlands. [4] Microsoft Quantum Lab Delft, GA Delft 2600, The Netherlands. ✉email: Sebastian.Heedt@Microsoft.com

**S**imilar to a light wave, an electron wave acquires a phase when interacting with a scattering centre. Studying this effect requires an interferometer with phase-coherent transport such as semiconducting or metallic rings[1–3]. In these nanostructures, the phase difference between the two paths ($\Delta\varphi$) can be tuned by a magnetic flux via the Aharonov–Bohm (AB) effect:

$$\Delta\varphi = 2\pi\frac{\Phi_B}{\Phi_0}, \qquad (1)$$

with $\Phi_B$ the magnetic flux through the interferometer and $\Phi_0 = h/e$ the flux quantum.

When the scattering centre is a quantum dot (QD), as depicted in Fig. 1a, the transmission phase $\varphi$ provides information complementary to the transmission probability $T = |t|^2$, with $t$ the transmission amplitude $t = \sqrt{T}e^{i\varphi}$. It can reveal insights into microscopic details of electron transport and into the spatial symmetries of the orbitals[4–7].

Recently, theoretical proposals suggested using interferometry as a read-out method of topological qubits, where quantum information is encoded in the electron parity of Majorana modes in semiconducting–superconducting nanowires[8–14]. Here, opposite qubit states are characterised by different transmission phases similar to the mesoscopic phase behaviour observed in few-electron QDs (Fig. 1b)[15–18]. When Majorana modes are absent, the phase is expected to exhibit the universal behaviour detected in many-electron QDs. In these systems, abrupt phase lapses break the simple parity-to-phase relation (Fig. 1c)[5,18–22].

Despite the critical application in topological qubits, the phase read-out of a QD in a nanowire interferometer has not been demonstrated yet. While pioneering works employed two-dimensional electron gases[18–20,22], here we take advantage of the recent advances in the growth of nanowire networks[23,24] and demonstrate interferometric read-out of a QD defined in a nanowire. Our findings provide crucial insights for future topological qubits based on hybrid one-dimensional nanowire systems.

## Results

**Cotunnelling AB interference.** Our device is shown in Fig. 2a and consists of a hashtag-shaped network of hexagonal InSb nanowires of high crystalline quality[24]. In the top-right arm, negative voltages ($V_{T1}$ and $V_{T2}$) on the top gates, T1 and T2, create two tunnel barriers that define an X-shaped QD (pink region). The voltage on the plunger gate PG ($V_{PG}$) tunes its electron occupation. Likewise, the transmission in the bottom-left branch—the reference arm—can be varied from pinch-off to the open regime by adjusting the voltage $V_{RG}$ on the reference gate (RG). The $p$-doped Si/SiO$_x$ substrate allows global back-gate (BG) functionality. A DC bias voltage with a small AC excitation, $V_{SD} + \delta V_{AC}$, is applied between source and drain, yielding a current $I + \delta I_{AC}$. Both the DC current and the differential conductance $G = \delta I_{AC}/\delta V_{AC}$ are measured in a dilution refrigerator with an electron temperature of $T_{el} \sim 35$ mK at its base temperature.

When the QD is not defined, the conductance at zero bias voltage displays AB oscillations as a function of the magnetic field perpendicular to the substrate ($B_\perp$) with period $\Delta B_\perp \sim 16-20$ mT

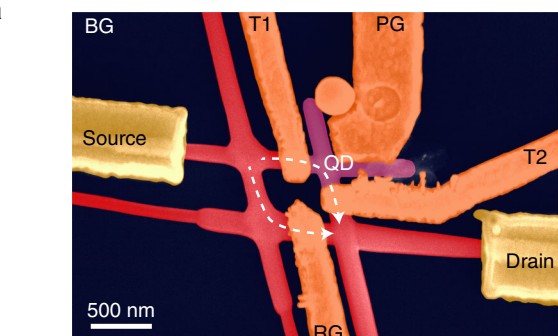

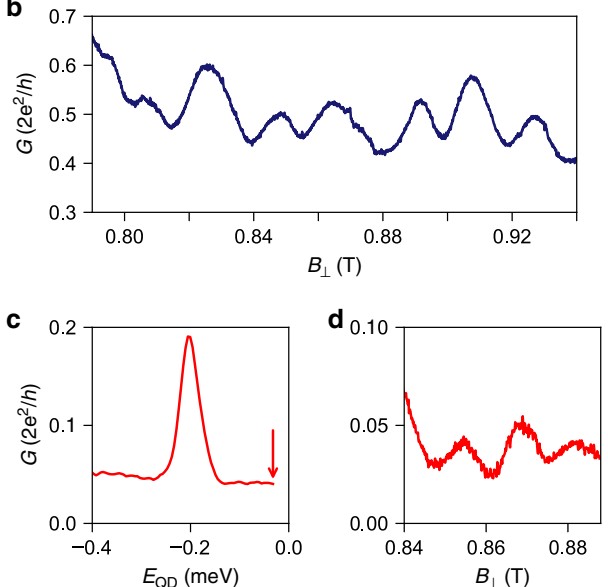

**Fig. 2 Aharonov–Bohm oscillations in an InSb nanowire network. a** False-colour scanning electron micrograph of the device: in red the nanowire network, in gold the leads, in orange the gates and in pink the quantum dot region. An additional illustration and a schematic of the device are shown in "Methods". **b** Conductance at zero bias voltage $G(V_{SD} = 0)$ as a function of the perpendicular field $B_\perp$ in the open regime (i.e., with no QD defined) manifesting AB oscillations. **c** $G(V_{SD} = 0)$ vs. $E_{QD}$ (the dot electrochemical potential) when the quantum dot is defined. **d** $G(V_{SD} = 0)$ vs. $B_\perp$ when the dot is in the cotunnelling regime (cf. Coulomb valley indicated by the red arrow in (**c**)).

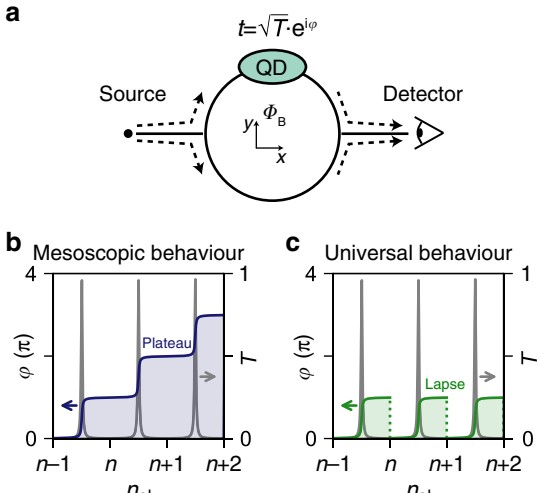

**Fig. 1 Mesoscopic and universal phase behaviours. a** The minimum setup to study the transmission phase via a quantum dot (in light green) is a two-path interferometer. **b**, **c** Transmission phase $\varphi$ and probability $T$ as a function of the electron number $n_{el}$ in a quantum dot. A Breit–Wigner function describes each of the resonances (in grey). **b** The mesoscopic regime: phase plateaus in the Coulomb valleys appear at 0 and $\pi$. **c** The universal regime: phase lapses occur between transmission resonances.

(Fig. 2b). This periodicity corresponds to a loop area of $\Phi_0/\Delta B_\perp \sim 0.21-0.26$ μm$^2$, which is consistent with the actual area of the device of $\sim 0.23$ μm$^2$ measured up to the centre of the nanowires.

When the QD is defined, we adjust the plunger-gate voltage between two resonances, as indicated by the red arrow in Fig. 2c, where the horizontal axis is the QD electrochemical potential ($E_{QD} = e \cdot \alpha \cdot V_{PG}$, with $\alpha$ the lever arm and $e$ the electron charge). In this regime, the electron–electron repulsion in the dot suppresses the current almost completely, which is known as Coulomb blockade. Transport is then allowed only via virtual, higher-order processes. At zero bias, elastic cotunnelling is predominant and its phase coherence is critical for parity-protected read-out schemes of Majorana wires[13,14].

When we balance the current distribution in the two arms of our device, the AB oscillations in the cotunnelling regime become visible with an amplitude of $\sim 20-30\%$ of the average conductance (Fig. 2d). The large visibility demonstrates that cotunnelling across the large Coulomb-blockaded dot is phase-coherent, fulfilling a fundamental requirement of future parity read-out circuits.

**From Coulomb to Fano resonances.** In order to characterize the QD, we first pinch off the reference arm. The green trace in Fig. 3a displays a series of nearly equally spaced conductance peaks stemming from tunnelling via the dot. Their separation is

also known as the addition energy and arises from two effects: the quantum confinement and the Coulomb interaction[25]. In a large dot, the second effect dominates over the first, leading to a series of peaks that are equidistant[25,26]. From the bias spectroscopy in Fig. 3b, we estimate the Coulomb charging energy $E_c = e^2/C \sim 0.35 - 0.45$ meV (with $C$ the overall capacitance) and the level spacing due to confinement $\delta \sim 0.020 - 0.035$ meV. We evaluate the first parameter from the size of the diamonds in bias voltage, and the second from the separation between the lines that confine the Coulomb diamonds and the lines that are due to the excited states. The large ratio of $E_c/\delta \gg 1$ indeed arises from the large size of the dot, which is designed to be comparable with the typical micron-long semiconducting–superconducting dots of near-future explorations[27–29]. Assuming a typical open-channel electron density of $2 \cdot 10^{17}$ cm$^{-3}$ [30] and the dot volume of $1.4 \cdot 10^{-2}$ μm$^3$, we estimate the maximum number of electrons on the QD to be $\sim 1 - 3 \cdot 10^3$.

We now start to activate transport in the reference arm. Upon increasing its transparency, the Coulomb peaks first evolve into the asymmetric peaks of the blue trace and then into the dips in the orange one of Fig. 3a. The variation of their line-shapes stems from the Fano effect, a phenomenon observed in multiple contexts in physics: from Raman scattering[31,32] to photon absorption in quantum-well structures[33,34], from transport in single-electron transistors[35] to AB interferometers[36–41].

The effect originates from the interference between two partial waves: one is undergoing a resonant scattering and the other is travelling through a continuum of states. In our experiment, the first is mediated by the discrete dot spectrum provided by Coulomb blockade and confinement, and the second by the continuum of the density of states in the reference path. Bias spectroscopy with the reference path being partially conducting—similarly to the blue trace in Fig. 3a—shows Fano peaks extending into the Coulomb valleys at $V_{SD} \sim 0$ mV (cf. black arrows in Fig. 3c). To the best of our knowledge, this is the first observation of Fano physics in a nanowire-based interferometer.

To distinguish the three regimes of Fig. 3a, we fit the line-shapes of the peaks using a generalized Fano model[37]. The relevant ingredients are the coupling terms between the dot and the two leads ($J_L$ and $J_R$), the transmission through the reference arm ($t_{ref}$) and the magnetic flux through the ring ($\Phi_B$). A schematic illustration and more information are shown in "Methods" and the result of the fits are listed in Supplementary Tables 1–3.

We extract the Fano parameter $F = t_{ref}/\sqrt{J_L J_R}$ from each peak (or dip). The inset of Fig. 3a shows that the averages of $F$ across each trace extend over three orders of magnitude, reflecting the large tunability of the device.

**The universal phase behaviour.** Upon sweeping the magnetic field, the Fano line-shapes vary periodically owing to the AB effect. In particular, Fig. 4a shows that two adjacent Fano resonances evolve in-phase.

We use the model described above to fit both peaks as a function of magnetic field, and we illustrate the result in Fig. 4b. The model captures well the main features of the experimental data, and the good agreement is visible in the line-cuts presented in Fig. 4c. Here, the three traces are taken at the positions denoted by the black, red, and green lines in both a and b. A $\pi$-shift in the AB oscillations is visible between both the black and red as well as the red and green traces. The complete evolution of the phase $\varphi$ as a function of $E_{QD}$ is extracted by tracking the maximum of the AB pattern and shown in the top panel of Fig. 4d. In the bottom panel, we present horizontal line-cuts of Figs. 4a and b at the positions indicated by the coloured lines.

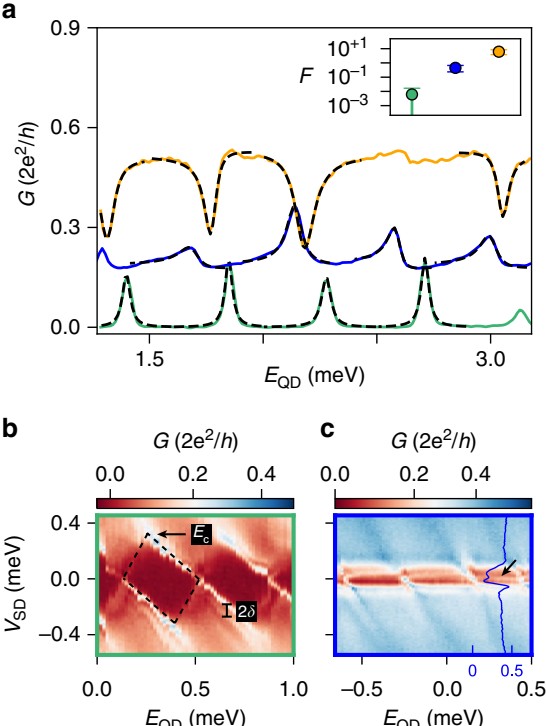

**Fig. 3 From Coulomb to Fano resonances. a** Differential conductance $G$ as a function of $E_{QD}$ with the reference arm fully pinched-off (green trace), partially conducting (blue trace) and transparent (orange trace). Dashed lines are best fits. Inset: Fano parameter $F = t_{ref}/\sqrt{J_L J_R}$, averaged across four peaks in each of the three regimes. **b, c** $G$ versus $E_{QD}$ and $V_{SD}$ in the first and second regime, respectively. The blue line-cut in (**c**) is taken at $E_{QD} = 0.32$ meV, the blue values on the horizontal axis refer to conductance $G$ in 2e$^2$/h. In (**b**), $E_c$ indicates the charging energy (at the apex of the diamond) and $\delta$ denotes the level spacing due to quantum confinement.

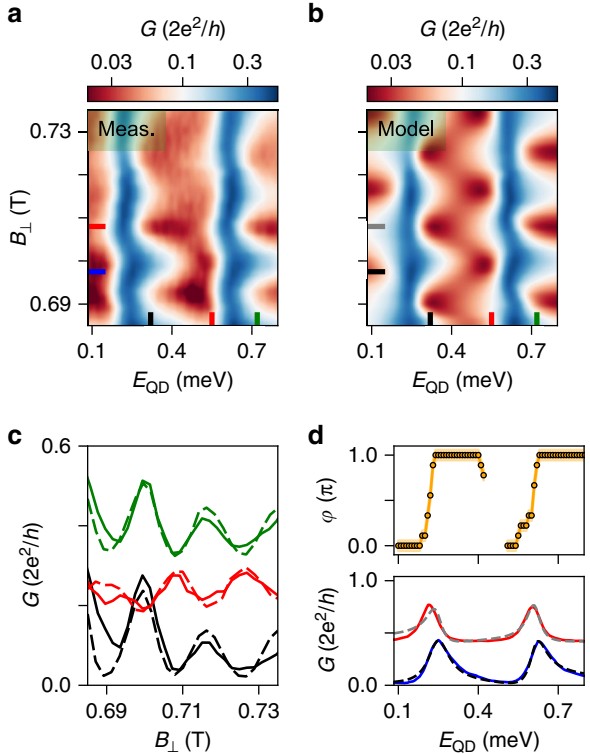

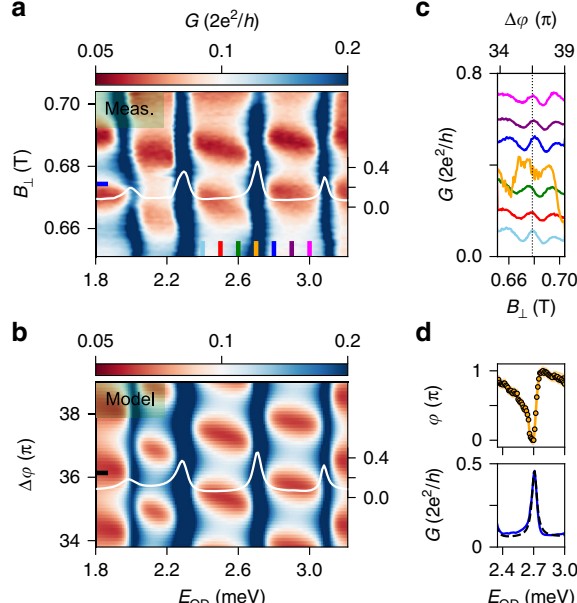

**Fig. 4 The universal phase behaviour. a** $G$ versus $E_{QD}$ and $B_\perp$ at zero bias voltage, describing the evolution of two adjacent Coulomb peaks as a function of magnetic field. Data are taken at back-gate voltage $V_{BG} = 1.5$ V. **b** Fit of the data in (**a**). **c** Solid and dashed traces are vertical line-cuts from (**a**) and (**b**), respectively, indicated by black, red, and green lines. The three pairs of curves are displaced by an offset of $0.15 \cdot 2e^2/h$ for clarity. **d** Bottom panel: solid and dashed lines are horizontal line-cuts of (**a**) and (**b**), respectively, at the positions indicated by the blue/black and red/grey lines. Top panel: transmission phase $\varphi$ extracted from the AB pattern. The shaded region indicates the error bars that stem from the uncertainty in extracting the oscillation maxima that is ~1−2 mT.

**Fig. 5 Multi-path transport effects. a** $G$ vs. $E_{QD}$ and $B_\perp$ exhibiting the evolution of four CPs. Data are taken at back-gate voltage $V_{BG} = -1.5$ V. **b** Calculated conductance assuming a multi-path interferometer, details are reported in the "Methods". The white traces in (**a**) and (**b**) correspond to the values indicated by the horizontal lines, and the vertical axes refer to conductance $G$ in $2e^2/h$. **c** Vertical line-cuts of (**a**) showing the evolution of AB oscillations across a charge transition in the QD. Traces are displaced by $0.1 \cdot 2e^2/h$ for clarity, except for the orange one taken on resonance. **d** Top panel: the trend of the AB maxima across the third CP. The shaded region indicates the error bars stemming from the uncertainty in extracting the oscillation maxima that is ~1−2 mT. Bottom panel: solid and dashed lines are horizontal line-cuts of (**a**) and (**b**), respectively, at the position indicated by the blue/black lines.

Here, we observe two main features: a phase variation of $\pi$ at the resonances over an energy scale similar to the broadening of the peaks, and a phase lapse in the Coulomb valley. These are distinctive features of the universal phase behaviour and are consistent with the in-phase evolution of the two adjacent CPs in Fig. 4a[5,18–21]. The observation of the universal rather than the mesoscopic behaviour can be explained by taking a look at the energy scales of the transport. In our measurement, the typical dot coupling energy ($\Gamma = \sqrt{|J_L J_R|} \sim 0.1$ meV) is a few times larger than the level spacing in the dot ($\delta \sim 0.02 - 0.035$ meV). Therefore, tunnelling occurs via multiple dot-levels, a condition for which theory predicts the observation of the universal behaviour[5,21].

Because previous experiments focused on the single-level regime ($\Gamma < \delta$)[18–20] and in the crossover ($\Gamma \sim \delta$)[22], finding both phase lapses and phase plateaus, our investigation in the fully multi-level regime ($\Gamma \gg \delta$) seems to complete the complex dot-interferometry puzzle.

**Multi-path transport effects.** In the following, we highlight that an optimal read-out of the transmission phase requires an interferometer close to the one-dimensional limit in the sense that it shoud comprise thin nanowires enclosing a relatively large hole.

For a gate configuration different from the previous regime, the transmission phase varies smoothly between several pairs of

adjacent CPs. Here, the phase displays a behaviour in between the universal and the mesoscopic regimes (Fig. 5a). The two configurations differ in the BG voltage that has been lowered from $V_{BG} = 1.5$ V in Fig. 4 to $V_{BG} = -1.5$ V in Fig. 5. Voltages on the tunnel gates are also re-adjusted to retain a similar transmission, whereas the plunger gate remains at $V_{PG} \sim 0$ V. We estimate a reduction of the electron density by no more than ~20% compared to the first case, leaving the dot still in the many-electron regime (see Supplementary Fig. 2).

In Fig. 5a, we show a colour map of $G$ vs. $E_{QD}$ and $B_\perp$, exhibiting the evolution of 4 CPs. The red features in the cotunnelling regions oscillate as a function of $B_\perp$ owing to the AB effect. Several vertical line-cuts are shown in Fig. 5c. The maxima of the AB oscillations around $B_\perp \sim 0.68$ T are converted into transmission phase via the magnetic field period (here $\Delta B_\perp = 19$ mT) and displayed in the top panel of Fig. 5d. In the bottom panel of Fig. 5d, we show a horizontal line-cut taken at the position indicated by the blue line in Fig. 5a. Similar to the data in Fig. 4, the phase exhibits a ~$\pi$ variation concomitant with the peak in the conductance. However, the phase lapse in the Coulomb valley is replaced by a smooth evolution. This slow phase variation is not universal, but depends on the specific gate setting.

We interpret this anomaly as a consequence of the relatively large width of the nanowires (~100–150 nm). Microscopically, we speculate that consecutive charge states might not couple to the same loop trajectory. The presence of at least two paths gives rise to beatings in the magneto-conductance that conceal the

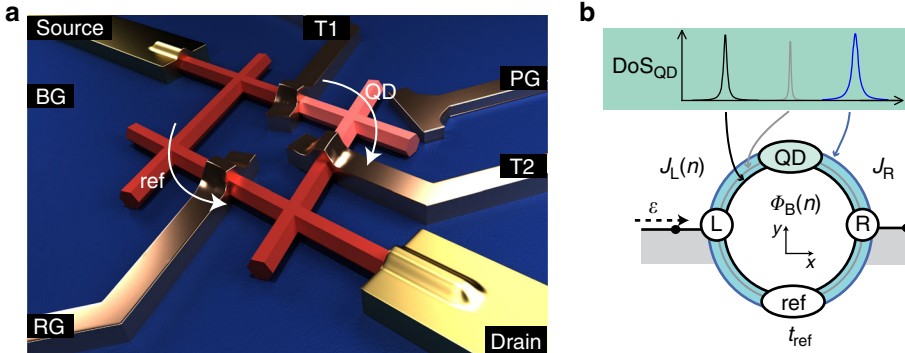

**Fig. 6 The multi-path interferometer. a** Illustration of the device: in red the nanowire, in gold the leads, in copper the gates and in pink the quantum dot. **b** Schematic of the device: the quantum dot exhibits a density of states (DoS$_{QD}$) comprised of discrete levels with distinct energy broadening. In the model, we assume that the dot states might couple to different interferometer trajectories that are sketched as rings of different colour.

evolution of the transmission phase[37]. Our interpretation is well-supported by the large width of the AB peak in the Fourier spectrum shown in Supplementary Fig. 3.

While in reality multiple trajectories could couple to each QD orbital, we reproduce our observation in the model by linking each resonance to a possibly different AB periodicity. This simple assumption enables to capture the main features of the measurement (Fig. 5b).

The coexistence of the two distinct phase behaviours (Fig. 4 vs. Fig. 5) in the same mesoscopic device is hard to fully explain, and might be correlated with the exact coupling mechanism between the dot orbitals and the leads.

## Discussion

In summary, we report interferometric measurements on a QD embedded in a network of four conjoint InSb nanowires. The observation of pronounced quantum interference in the cotunnelling regime and the presence of Fano resonances suggest that interferometry is a viable tool for parity read-out of future topological qubits in nanowire networks. Theory suggests that the transmission probability of a semiconducting–superconducting QD in the topological regime should exhibit phase plateaus[15,16]. However, transmitting channels other than the teleportation via Majorana bound states were not taken into account. In experiments, extended topologically trivial modes without an underlying topological bulk phase can mimic Majoranas. Hence, quasiparticle transport via these modes might offer parallel paths to the Majorana teleportation[42,43]. Altogether, these can cause phase lapses that hinder the simple correspondence between the transmission phase and the electron parity. We conclude by remarking that future interferometers for parity-state discrimination via phase read-out should be designed with a large ratio between circumference and nanowire diameter.

## Methods

**Device fabrication.** InSb networks are grown by combining the bottom-up synthesis of four monocrystalline nanowires and the accurate positioning of the nanowire seeds along trenches on an InP substrate. Further details on the nanowire growth are presented in refs. [23] and [24].

After the growth, we transfer nanowire networks from the InP growth chip onto a $p$-doped Si/SiO$_x$ substrate (oxide thickness of 285 nm) using a mechanical nanomanipulator installed in a scanning electron microscope. Ti/Au contact leads are patterned using electron-beam lithography and e-gun evaporation, following surface treatment of the InSb for 30 min in a sulfur-rich ammonium polysulfide solution diluted in water (1:200) at 60 °C. The devices are covered with ~30 nm of sputtered Si$_x$N$_y$ acting as a gate dielectric. The second layer of Ti/Au electrodes is patterned and evaporated to define the top gates. The chip is then diced, mounted and bonded onto a commercial printed circuit board.

**Transport measurements.** The device is cooled down in a dry dilution refrigerator equipped with a 6-2-2 T vector magnet. The base electron temperature is $T_{el} \sim 35$ mK. Conductance across the device is measured via a standard low-frequency lock-in technique at an AC signal amplitude of $\delta V_{AC} \sim 20$ μV. The data presented in the main text and in Supplementary Figs. 2, 3, and 4 are taken from a single device. In Supplementary Figs. 5 and 6, we present data taken from a second and third device, respectively. The AC conductance in Figs. 2, 3, 4, and Supplementary Fig. 4 was corrected for a constant offset that was later identified to arise from the setup.

**Model of the AB interferometer.** The Landauer formula ($G = (2e^2/h) \cdot T$) connects the single-channel conductance of the system with the transmission probability $T$. In Fig. 6, we show a schematic of the multi-path AB interferometer (a simple generalization of the single-path counterpart) next to an illustration of the actual device. The QD electrochemical potential ladder is represented as a series of discrete states, separated by the charging energy $E_c = e^2/C$[26]. For the single-path case, hopping terms $J_L$ and $J_R$ couple the source (L) and drain (R) to the QD, respectively, with the AB phase included in $J_L = j_L \cdot \exp(i2\pi\Phi_B/\Phi_0)$, and $j_L$ and $J_R$ being real parameters. $\Phi_B$ is the magnetic flux through the loop. When multiple-path are considered, we define the phase of $J_L$ as $2\pi\Phi_B/\Phi_0[1 + x(n)]$, with the parameter $x(n)$ distinct for every CP.

The reference site has a slowly varying spectrum that we will assume for simplicity to be constant. The leads are assumed to be one-dimensional (lattice constant $a$) with hopping matrix elements $-J$ and a typical energy dispersion of $\varepsilon = -2J\cos(ka)$. The resulting transmission probability $T$ through the AB interferometer is[37]

$$T = \frac{4|S_{LR}|^2 \cdot \sin^2(ka)}{\left||S_{LR}|^2 - (S_{LL} + e^{-ika})(S_{RR} + e^{-ika})\right|^2},$$ (2)

with

$$S_{XY} = \sum_{n=0}^{N} \frac{J_X(n)J_Y(n)^*}{J(\varepsilon - E_{QD} + E_n)} + t_{ref},$$ (3)

where $E_n$ represent the positions of the $N$ levels relevant in the tunnelling process on the $E_{QD}$-axis. For the fit of our results, we add an offset to Eq. (2) to capture the incoherent contribution of the current through the device.

## Data availability

The data that support the findings of this study are available at https://doi.org/10.4121/uuid:9e625b55-11cf-4de2-8b81-32b5bf04d53d.

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

## Acknowledgements

We gratefully acknowledge Bernard van Heck for fruitful discussions and we thank Alexandra Fursina and Christine Nebel for the growth substrate preparation. This work has been supported by the European Research Council (ERC HELENA 617256 and Synergy), the Dutch Organization for Scientific Research (NWO) and Microsoft Corporation Station Q.

## Author contributions

F.B., K.Z. and S.H. fabricated the devices. F.B., S.H. and K.Z. performed the measurements. F.B. analysed the transport data. F.B., S.H. and L.P.K. discussed the results. S.G. and R.L.M.O.h.V. carried out the growth and the transfer of the interconnected InSb nanowires under the supervision of E.P.A.M.B. F.B. wrote the manuscript with contributions from S.H. and all authors provided critical feedback. S.H. and L.P.K. supervised the project.

## Competing interests

The authors declare no competing interests.
