## [Peer Review File · Nature Communications]

Reviewers' comments:

Reviewer #1 (Remarks to the Author):

This paper studies the Aharonov-Bohm effect in semiconducting-nanowire-based interferometers. In particular, the focus of the paper is on detection of the transmission phase through a network of interconnected InSb nanowires when one of the arms contains a quantum dot. In my opinion, the reading is somewhat intricate, and the presentation is rather technical. While the presented results might be of interest for the mesoscopic physics community, I find that the manuscript lacks the character of generality and clarity that is necessary for the readership of Nature Communications. I understand that these results are motivated by the prospect of using interferometry for read-out of topological qubits (through the dependence of the transmission phase on electron parity), but, as it is now, this manuscript just presents a detailed analysis of how two distinct quantum dot regimes (few-electron and many electron) affect the transmission phase through the interferometer. Concerning novelty, there are various previous papers reporting on this kind of experiments (Refs. 14-17), although not using nanowire networks. One could argue that showing interference effects and quantum coherence through such networks is already an important achievement (which is true), but this has been already reported by the same group in Ref. 23. In what follows, I explain this overall assessment by asking/pointing out specific questions/points that should be addressed:

1) Having in mind that this is an experimental paper, I find it very strange that the authors stress the importance of their tight-binding model (even at the abstract level) in order to understand their results. As it is clear when reading the manuscript, the physics described here is just that of Fano interference between discrete resonances (the Coulomb blockade peak) and the contribution from the continuum. In this respect, the phenomenon is general and not limited to the specific modeling (here tight binding) of the transmission (Eqs. 2 and 3). In my opinion, the manuscript would have benefited from a more general explanation in terms of Fano interference.

2) As I mentioned in my general assessment, the paper is rather technical. This happens right from the start, where the authors talk about read-out of topological states to motivate their experiments. Why parity read out is such a big deal? (this is a rhetorical question: I know why parity read out is a big deal but not the readers necessarily...). This is also a bit confusing since, as far as I can see, they don't clearly demonstrate parity-dependent readout in a mesoscopic regime in their device. More technicalities can be found at the beginning of the paper (the distinction between two- and four-terminal configurations, the breakdown of Onsager relations, phase rigidity, etc) with almost no context/explanation. This continues in the next paragraph which is plagued with technicalities: spin-orbit and its relation with topological qubits, cotunneling regime, tight-binding model...

3) When discussing the results (section II), I would suggest to use an image of the actual sample (the one shown in Fig 6) together with the illustration. Also, it would be very useful at this level of the paper to find a minimal explanation of what a Coulomb blockade peak/valley is and why cotunneling measurements are important.

4) Related to the previous point: their model is just a tight-binding chain coupled to N resonances. Is Coulomb blockade irrelevant to understand the main mechanism of Fano interference? If yes, the authors should try to present their results/explanation in a more general language, avoiding jargon as much as possible.

5) The universal behaviour through phase lapses is clearly demonstrated in these experiments (and, as claimed by the authors, this is probably the first time that this regime has been studied in detail). Unfortunately, this is not the relevant regime towards read out of topological qubits (since in the

universal regime, phase lapses are not linked to the electron number). Here, some numbers are mentioned (like the full width half max, the level spacing, etc), but it would be nice to see a short explanation of what these parameters are (even at a pictorial level) when these numbers are first mentioned in the previous paragraph (i.e the paragraph describing how they are extracted when the reference arm is pinched off).

6) Section D aims at understanding the quasi-1D limit (relevant for Majorana wires). Here the effect of complicated multi path trajectories is elucidated which results in a phase behaviour somewhat in between mesoscopic and universal. In my view, this is probably the more relevant section of the paper since this is probably the first experiments where the role of topologically-trivial modes on the so-called Majorana teleportation (and their specific contribution to transmission phase lapses) is discussed. In this context, I like also section III (Discussion) since the implications of this data set are discussed in a more general (and relevant) context.

Given the above arguments, I do not recommend the paper for publication in Nature Communications. Some parts of the paper, see my point 6 above, might be considered of enough relevance but, as it is now, the manuscript does not stress enough the novelty and/or relevance of the experiments presented here (even at the abstract level this last part about multiple trajectories is not highlighted enough). My recommendation would thus be to considerably rewrite the manuscript given my above criticism. Importantly, the sentence in the discussion part "Nevertheless, we conclude by remarking that, despite the presence of phase lapses, the transmission phase remains parity-sensitive in the vicinity of the transmission resonances." should be carefully explained (and demonstrated by measurements). A detailed and convincing explanation of experimental data demonstrating deviations from the simplistic theory presented in Refs. 11-12 could be reconsidered for publication in Nature Communications.

Reviewer #2 (Remarks to the Author):

The authors demonstrate an electron interferometer integrating a quantum dot in one of its arms. The study is performed on devices built starting from crossed-wire InSb nanostructures [23] and can be relevant in view of phase-resolved tunnel spectroscopy of Majorana states. The work specifically focuses on the evolution of tunneling phase as a function of the electron filling in the quantum dot, depending on a variety of parameters. Part of the results can be framed within the so-called "universal behavior", which was demonstrated at Weitzmann in 2005. The main novelties of the paper are:

(i) the physical system used to implement the interferometer. Crossed InSb nanowires are used here; this is a rather unique system, and one of the election systems for the creation of Majorana states for topological quantum computation.

(ii) the devices work in a multi-level regime, which is unusual. Results in the literature have been typically reported in the single-level regime, i.e. in the quantum Coulomb blockade limit. Here we are apparently in a cross-over towards the classical Coulomb blockade limit.

I believe the manuscript is interesting and related to a timely research subject. Devices and measurements are at the state of the art. Some of the interpretations are a bit speculative but in my view the paper is innovative enough to be suitable for Nature Comm. I believe some points should be clarified/fixed before publication:

1. I have few comments on the presentation. One concerns figure 3a and the DoS sketched therein. First question: the authors report markedly different lineshapes, which suggest - I guess - that different orbitals have markedly different hopping parameters. Is this an evidence, an expectation, or? I think this might be better clarified. Second question: the authors use a 1D TB model, but I see here a 3D-like DoS. I would expect the true DoS of the InSb conductor to be indeed close to 3D, but a clarification is maybe needed. Also, I wonder if the functional form of the DoS is really relevant here. I would guess that the InSb conductor has an almost "metallic" DoS with mesoscopic fluctuations on top? I also note that in other experiments 1D indeed appeared to be a more appropriate description. Since the existence of multiple areas and trajectories in the ring is relevant at the end of the paper, I would maybe spend few words on the issue, in the initial part of the paper.

A second (really minor) comment concerns "N" after equation (3). I would clarify that N is the number of levels relevant to the tunneling process. I think an average reader might not expect that many levels participate to the transport. The authors mention that cotunneling is important in the experiment, but I think this does not necessarily imply that $\Gamma > \Delta$.

2. Concerning the phase evolution shown in Fig.4d, I see there are some missing points due - I guess - to the difficulty/impossibility in performing a reasonable fit of the data. I am sure that this is not an on/off problem... can the authors maybe add some error bars so that the transition from a well-defined phase to an unknown phase is better quantified? I also notice that the 4kBT scale bar is a not very easy to understand, possibly because of its low aspect ratio it does not look like a scale bar.

Moving to the physics, I notice that the "smooth" phase transition from 0 to π actually occurs on an energy scale on the order of Γ . Is this just a coincidence? I am wondering if it is reasonable to expect a transition width $\propto kBT$ in the regime studied by the authors, or if rather Γ is the important scale. I am not aware of any theoretical prediction, my observation comes from a naive analogy with what happens to the CP linewidth ($\propto kBT$ in the single-level regime, $\propto \Gamma$ when lifetime broadening dominates). The discussion about the breakdown of time-reversal is interesting but its connection with the effect remains a bit vague. Also, are lower magnetic field data available maybe?

3. The different phenomenology presented in D is intriguing, but also puzzling. I appreciate the fact that the authors were able to reproduce the effect by adding some complexity to their model . However, in the present version of the paper the transition from evidences in C to those in D remains a bit obscure to me and one has the impression that fine adjustment has been introduced in the modelling. I don't want to be negative, I think the evidence is interesting and deserves publication. I would at least report the fit parameters somewhere and try and give some intuitive explanation of the mechanism leading to the change in the phase pattern.

4. Datasets in C and D refer to $V_{BG}=+1.5V$ and $-1.5V$, respectively. I thus expect to have less electrons in D? Why is area less well-defined in the D configuration? Since there is less carrier density, I would expect the ring area to be MORE well-defined in configuration D than in C.

Reviewer #3 (Remarks to the Author):

Review report - NCOMMS-19-40640

In this manuscript, the authors study the transmission phase of a quantum dot (QD) in the many-electron regime via interferometry. The experimental system consists of a network of four InSb nanowires arranged in a hashtag shape during epitaxial growth, acting as a two terminal Aharonov-Bohm (AB) interferometer, where the gate defined QD is incorporated in one of the arms. Several

gates are employed to tune the carrier concentration in both the QD and the nanowire leads. The authors show that phase coherent transport is conserved through the QD in the Coulomb blockade regime as indicated by the observation of AB oscillations in the differential conductance (G) [Fig. 2]. A tight-binding model, using an expression for the total transmission from ref. [25], is used to fit the resonant peaks (dips) in G [Fig.3]. Furthermore, by tuning the gates, two regimes are identified: one with universal phase behavior (rapid phase changes of π close to the resonance followed by a phase laps in the Coulomb blocked region) [Fig.4]. The second regime, exhibit phase behavior in between the universal and the mesoscopic regime [Fig. 5]. This behavior is attributed to transport channels in the arms with different loop areas coupling to the different QD states.

This appears to be the first AB interferometry study on the InSb material system when a QD is included in one of the arms. Owing to the strong spin-orbit interaction and large g -factor, InSb is a promising material system to realize topological quantum computing. AB interferometry including a large "topological QD" has been suggested as a mean for read-out of topological qubits, but also for distinguishing between trivial and topological states in semiconductor-superconductor hybrid devices which is an important step towards topological quantum computing.

The manuscript is well-organized and easy to follow. I appreciate the clear experimental data and the close resemblance between experimental and model results. Thus, overall I recommend this paper for publication in Nature communications given that the comments below are addressed:

(1) The authors point out the particular regime studied where multiple orbitals on the QD contribute to the transport has never been studied before. Can the authors comment on why this is an interesting regime to study? How would the effect of multiple lead channels masking the intrinsic transmission phase affect read out of topological qubits? Is this effect robust when lowering the back gate further? Can the authors comment on electron density in the studied regime vs the electron density needed for entering topological regime in a semiconductor-superconductor hybrid device?

(2) Fig. 2 caption: Define " G ".

(3) Regarding the definition of the "QD electrochemical potential": $E_{\text{QD}} = \alpha V_{\text{PG}}$. Strictly speaking, this is only the gate contribution to the electrochemical potential. Is there an elementary charge missing or how do you define α ? Not $C_{\text{gate}}/C_{\text{total}}$?

(4) Regarding Fig. 3c caption: E_c and Δ are not defined in the caption. To me it looks like the bar indicating Δ is shifted slightly too far down (I assume it is supposed to indicate the distance between the conductance line of the border of the diamond and the first parallel line). In addition, the indicated distance is 2 times the orbital spacing whereas in main text Δ is referred to as the orbital spacing. How has the orbital spacing been extracted? Furthermore, the dashed line indicating the diamond seems to be slightly off, overestimating the E_c .

(5) Regarding estimation of electron population on the dot: For more transparency declare the estimated volume of the dot.

(6) Fig. 4: Add information on the DC bias in the caption. "magnetic flux" should be magnetic flux density of B-field. Expand the description of panel d (top) in the caption. From the caption, it is not clear why the $4k_{\text{BT}}$ marker is there.

(7) "In order to convert between the evolving phase of $J_L = j_L \exp(i2 \Phi)$ and the magnetic field we use the area of the loop." Can you be more specific here and insert the full relation? Furthermore, can you include a more detailed description how you extract the transmission phase in Fig. 4d (or refer to the SI)?

(8) Regarding the SI: I suggest to refer to the SI in the main text to highlight how the SI is supporting the main text?

(9) Can you tune the QD into the few electron regime or have you a device with smaller distance between the defining gates? If so, can you see mesoscopic phase behavior or is the multiple channel effect hindering to see the mesoscopic phase behavior?

(10) Regarding the last sentence in the conclusion: "Nevertheless, we conclude by remarking that, despite the presence of phase lapses, the transmission phase remains parity-sensitive in the vicinity of the transmission resonances. What do you mean with parity sensitive? That you see a π -shift at the resonance (change of parity)? But you cannot extract the absolute parity of the dot from the transition phase?"

Response to the reviewers' comments:

We thank the referees for taking the time to consider our paper in detail. We appreciate their critical review and have substantially rewritten our manuscript accordingly. We show below in black the remarks from the referees, in blue our response. In the revised manuscript (and in the Supporting Information) the amendments are highlighted in orange.

Reviewer #1 (Remarks to the Author):

This paper studies the Aharonov-Bohm effect in semiconducting-nanowire-based interferometers. In particular, the focus of the paper is on detection of the transmission phase through a network of interconnected InSb nanowires when one of the arms contains a quantum dot. In my opinion, the reading is somewhat intricate, and the presentation is rather technical. While the presented results might be of interest for the mesoscopic physics community, I find that the manuscript lacks the character of generality and clarity that is necessary for the readership of Nature Communications. I understand that these results are motivated by the prospect of using interferometry for read-out of topological qubits (through the dependence of the transmission phase on electron parity), but, as it is now, this manuscript just presents a detailed analysis of how two distinct quantum dot regimes (few-electron and many electron) affect the transmission phase through the interferometer. Concerning novelty, there are various previous papers reporting on this kind of experiments (Refs. 14-17), although not using nanowire networks. One could argue that showing interference effects and quantum coherence through such networks is already an important achievement (which is true), but this has been already reported by the same group in Ref. 23.

We thank the referee for bringing up these points.

In our second submission, we have addressed all these concerns. In fact,

- we have changed the abstract entirely, such that it now provides a more general picture of the content of our manuscript;
- we have substantially reduced the number of technicalities in favour of legibility, clarity and a more instructive presentation;
- we have moved the relevant technicalities to the Supporting Information;
- we have emphasised the connection between our experiment and the context of Majorana physics;
- we have presented the Fano phenomenon in a very general way in section II B (that has now a new title 'From Coulomb to Fano resonances');
- we have moved the details regarding the theoretical model from section II B, C and D into the Methods;
- we have adjusted Figures 2, 3, 4 and 5 and the captions according to the proposed modifications;
- we have stressed the novelties of our work by:
 - emphasising in the abstract and in the main text that dot-interferometry has been reported so far only in two-dimensional electron gases;
 - clarifying that the study of a large quantum dot is motivated by topological qubit proposals;
 - stating that parity-conserving qubits require read-out in the cotunnelling regime via coherent virtual processes;
 - stressing the novelty of the Fano effect, not reported yet in nanowire-based devices;
 - highlighting that the multi-level regime (with level spacing much smaller than dot-lead coupling) was not investigated before. Previous works focused on the single-level and the

crossover regimes (Yacoby et al., PRL 74, 1995; Schuster et al., Nature 385, 1997; Avinun-Kalish et al., Nature 436, 2005; Edlbauer et al., Nat. Commun. 8, 2017);

- emphasising the results of section II D that clearly demonstrate how the transmission phase is not clearly detectable when multiple paths are present in Aharonov-Bohm interferometers;
- concluding with the remark that future interferometers for parity read-out should be realized in a way that avoids the multi-path regime.

Altogether, these aspects make our work substantially different from Gazibegovic et al., Nature 584, 2017. In fact, we utilise the nano-networks grown with the method from Gazibegovic et al. to establish interferometry measurements on a nanowire-based quantum dot and, with a critical spirit, we conclude by pointing out the limitations of this platform (i.e. the presence of multi-path effects).

In what follows, I explain this overall assessment by asking/pointing out specific questions/points that should be addressed:

1) Having in mind that this is an experimental paper, I find it very strange that the authors stress the importance of their tight-binding model (even at the abstract level) in order to understand their results. As it is clear when reading the manuscript, the physics described here is just that of Fano interference between discrete resonances (the Coulomb blockade peak) and the contribution from the continuum. In this respect, the phenomenon is general and not limited to the specific modeling (here tight binding) of the transmission (Eqs. 2 and 3). In my opinion, the manuscript would have benefited from a more general explanation in terms of Fano interference.

We really appreciate this remark. In this regard, we rewrote our discussion on the Fano effect. We have emphasised the generality of the phenomenon both in the abstract and in section II B, discussing other contexts where it manifests itself. To highlight our experimental data, we moved the intricate introduction of the model together with the equations to the Methods, leaving merely the basic description of the Fano parameter in section II B.

2) As I mentioned in my general assessment, the paper is rather technical. This happens right from the start, where the authors talk about read-out of topological states to motivate their experiments. Why parity read out is such a big deal? (this is a rhetorical question: I know why parity read out is a big deal but not the readers necessarily...). This is also a bit confusing since, as far as I can see, they don't clearly demonstrate parity-dependent read-out in a mesoscopic regime in their device. More technicalities can be found at the beginning of the paper (the distinction between two- and four-terminal configurations, the breakdown of Onsager relations, phase rigidity, etc) with almost no context/explanation. This continues in the next paragraph which is plagued with technicalities: spin-orbit and its relation with topological qubits, cotunneling regime, tight-binding model...

Based on the referee's concern, we have seriously reconsidered our presentation. We have now clarified and extended the text regarding the relationship between our experiment and the study of topological states. We have removed all the technicalities not immediately relevant to the storyline and, with the same purpose, moved the details related to the fits to the Methods. The technical discussion of the breakdown of the Onsager relation in the context of phase rigidity has been moved and expanded in the Supporting Information.

3) When discussing the results (section II), I would suggest to use an image of the actual sample (the one shown in Fig 6) together with the illustration. Also, it would be very useful at this level of the paper to find a minimal explanation of what a Coulomb blockade peak/valley is and why cotunneling measurements are important.

We replaced the illustration with a false-colour scanning electron microscopy of the device. We added a phenomenological description of Coulomb blockade physics in both sections II A and B, and better linked our data in the cotunnelling regime with the need to perform read-out of prospective qubits in the Coulomb blockade (see second to last paragraph of section II A).

4) Related to the previous point: their model is just a tight-binding chain coupled to N resonances. Is Coulomb blockade irrelevant to understand the main mechanism of Fano interference? If yes, the authors should try to present their results/explanation in a more general language, avoiding jargon as much as possible.

We thank the referee for pointing this out. Indeed, the Fano effect is a general phenomenon (Huang et al., AIP Adv. 5, 2015), therefore we added a sentence in section II B that illustrates the various contexts in which the Fano effect has been observed. The role of Coulomb blockade in the context of dot-interferometry was not been treated in detail so far. Previous works (Yacoby et al., PRL 74, 1995; Schuster et al., Nature 385, 1997; Avinun-Kalish et al., Nature 436, 2005; Edlbauer et al., Nat. Commun. 8, 2017) mainly focused on resonant transport via single dot levels neglecting the presence of Coulomb repulsion. In Aharony et al., Phys. Rev. B 73, 2006, the Coulomb repulsion energy is simply included in the spacing between the resonances, and we decided to adopt the same approach. Our experiment - conducted in the fully Coulomb-blockade regime - seems to confirm that the Fano effect does not depend on the exact origin of the transmission resonances (mainly Coulomb repulsion in our case vs. quantum confinement in previous works). We added two sentences in section II B to better illustrate this and avoid jargon altogether.

5) The universal behaviour through phase lapses is clearly demonstrated in these experiments (and, as claimed by the authors, this is probably the first time that this regime has been studied in detail). Unfortunately, this is not the relevant regime towards read out of topological qubits (since in the universal regime, phase lapses are not linked to the electron number).

The main goal of our work is to establish interferometric read-out of a large quantum dot in a nanowire-based circuit. This was made possible by the breakthrough of Gazibegovic et al., Nature 584, 2017 in growing epitaxial nanowire networks. Attempting to explore the relationship between the transmission phase and the Majorana parity requires a more advanced device, with the dot in contact with a parity-preserving superconductor such as Al, Sn etc. (L. Fu, Phys. Rev. Lett. 104, 2010). At the time of our experiment, this was beyond state-of-the-art nano-fabrication capabilities.

Our choice of quantum dot size had a large impact on the interferometry ‘working point’. In fact, with increasing the dot size, the dot level spacing becomes smaller, and tunnelling involving multiple-levels becomes more favourable than via single-levels. Altogether, this led to the observation of phase lapses (Oreg et al., New J. Phys. 9, 2007).

While our results differ from the case expected for a hybrid dot with Majorana states (L. Fu, Phys. Rev. Lett. 104, 2010; Drukier et al., Phys. Rev. B 98, 2018) or for a few-electron dot (Avinun-Kalish et al., Nature 436, 2005), understanding the universal phase behaviour is still important and highly relevant for qubit read-out. In fact, the ‘phase diagram’ (topological energy gap vs. chemical potential and Zeeman energy) predicted for semiconducting-superconducting nanowires is largely dominated by the topologically trivial regime, where the universal behaviour is expected (Winkler et al., Phys. Rev. B 99, 2018, Vaitiekėnas et al., Science 367, 2020; Antipov et al., Phys. Rev. X 8, 2018; de Moor et al., New J. Phys. 20, 2018). In addition, in a recent experiment (Whiticar et al., ArXiv 1902.07085, 2019) interferometry of a hybrid dot in a two-dimensional platform was performed, finding both phase plateaus and lapses without a clear interpretation.

Here, some numbers are mentioned (like the full width half max, the level spacing, etc), but it would be nice to see a short explanation of what these parameters are (even at a pictorial level) when these numbers are first mentioned in the previous paragraph (i.e. the paragraph describing how they are extracted when the reference arm is pinched off).

As suggested, we have added an explanation on the origin of level spacing and of the charging energy. In addition, we added a sentence in section II B that clearly states how they are evaluated.

6) Section D aims at understanding the quasi-1D limit (relevant for Majorana wires). Here the effect of complicated multi path trajectories is elucidated which results in a phase behaviour somewhat in between mesoscopic and universal. In my view, this is probably the more relevant section of the paper since this is probably the first experiments where the role of topologically-trivial modes on the so-called Majorana teleportation (and their specific contribution to transmission phase lapses) is discussed. In this context, I like also section III (Discussion) since the implications of this data set are discussed in a more general (and relevant) context.

We thank the referee for the appreciative remarks.

Given the above arguments, I do not recommend the paper for publication in Nature Communications. Some parts of the paper, see my point 6 above, might be considered of enough relevance but, as it is now, the manuscript does not stress enough the novelty and/or relevance of the experiments presented here (even at the abstract level this last part about multiple trajectories is not highlighted enough). My recommendation would thus be to considerably rewrite the manuscript given my above criticism. Importantly, the sentence in the discussion part "Nevertheless, we conclude by remarking that, despite the presence of phase lapses, the transmission phase remains parity-sensitive in the vicinity of the transmission resonances." should be carefully explained (and demonstrated by measurements). A detailed and convincing explanation of experimental data demonstrating deviations from the simplistic theory presented in Refs. 11-12 could be reconsidered for publication in Nature Communications.

Reviewer #2 (Remarks to the Author):

The authors demonstrate an electron interferometer integrating a quantum dot in one of its arms. The study is performed on devices built starting from crossed-wire InSb nanostructures [23] and can be relevant in view of phase-resolved tunnel spectroscopy of Majorana states. The work specifically focuses on the evolution of tunneling phase as a function of the electron filling in the quantum dot, depending on a variety of parameters. Part of the results can be framed within the so-called "universal behavior", which was demonstrated at Weitzmann in 2005. The main novelties of the paper are:

(i) the physical system used to implement the interferometer. Crossed InSb nanowires are used here; this is a rather unique system, and one of the election systems for the creation of Majorana states for topological quantum computation.

(ii) the devices work in a multi-level regime, which is unusual. Results in the literature have been typically reported in the single-level regime, i.e. in the quantum Coulomb blockade limit. Here we are apparently in a cross-over towards the classical Coulomb blockade limit.

I believe the manuscript is interesting and related to a timely research subject. Devices and measurements are at the state of the art. Some of the interpretations are a bit speculative but in my view the paper is innovative enough to be suitable for Nature Comm. I believe some points should be clarified/fixed before publication:

We thank the referee for the detailed review of our article and the positive verdict on this work.

1. I have few comments on the presentation. One concerns figure 3a and the DoS sketched therein. First question: the authors report markedly different lineshapes, which suggest - I guess - that different orbitals have markedly different hopping parameters. Is this an evidence, an expectation, or? I think this might be better clarified.

Semiconducting quantum dots usually exhibits levels with various broadenings which cause a non-uniform amplitude of the Coulomb resonances (see for instance our measurements in both regimes in Fig. 3 and Fig. 4 and the review Kouwenhoven et al., Rep. Prog. Phys. 64, 2001). We have moved the schematic of the model to the Methods and added a descriptive sentence in the caption of Fig. 6 covering this aspect.

Second question: the authors use a 1D TB model, but I see here a 3D-like DoS. I would expect the true DoS of the InSb conductor to be indeed close to 3D, but a clarification is maybe needed. Also, I wonder if the functional form of the DoS is really relevant here. I would guess that the InSb conductor has an almost "metallic" DoS with mesoscopic fluctuations on top? I also note that in other experimens 1D indeed appeared to be a more appropriate description. Since the existence of multiple areas and trajectories in the ring is relevant at the end of the paper, I would maybe spend few words on the issue, in the initial part of the paper.

We agree with this assessment. Considering the additional remarks by the other reviewers, we have decided to move the description of the model to the Methods. We have also decided to remove the sketch of the density of states of the reference arm, that – depending on the value of the Fermi wavelength and on the wire diameter – takes different forms (i.e. varying from 1D-like to 3D-like). The exact density of states in the reference arm is not relevant to the reader. Therefore, we have decided to emphasise the notion that the

reference arm has more generally and simply a continuum of states.

A second (really minor) comment concerns "N" after equation (3). I would clarify that N is the number of levels relevant to the tunneling process. I think an average reader might not expect that many levels participate to the transport.

We have added this note in the main text.

The authors mention that cotunneling is important in the experiment, but I think this does not necessarily imply that $\Gamma > \delta$.

We agree with the reviewer. This was not implied in the text.

2. Concerning the phase evolution shown in Fig.4d, I see there are some missing points due - I guess - to the difficulty/impossibility in performing a reasonable fit of the data. I am sure that this is not an on/off problem... can the authors maybe add some error bars so that the transition from a well-defined phase to an unknown phase is better quantified? I also notice that the 4kBT scale bar is a not very easy to understand, possibly because of its low aspect ratio it does not look like a scale bar.

We thank the referee for this consideration. Indeed, the missing points are due to the weakness of the Aharonov-Bohm oscillation at the centre of the Coulomb valley, which did not allow to estimate the transmission phase. In Figs. 4d and 5d, we decreased the size of the markers and coloured the region between the lines transitioning through the upper and lower limits of the error bars. The error bars of the transmission phase originate from a 1-2 mT uncertainty in defining the maximum of the Aharonov-Bohm oscillations. This information has been added to the captions of Figs. 4 and 5.

Moving to the physics, I notice that the "smooth" phase transition from 0 to π actually occurs on an energy scale on the order of Γ . Is this just a coincidence? I am wondering if it is reasonable to expect a transition width $\propto k_B T$ in the regime studied by the authors, or if rather Γ is the important scale. I am not aware of any theoretical prediction, my observation comes from a naive analogy with what happens to the CP linewidth ($\propto k_B T$ in the single-level regime, $\propto \Gamma$ when lifetime broadening dominates). The discussion about the breakdown of time-reversal is interesting but its connection with the effect remains a bit vague. Also, are lower magnetic field data available maybe?

In Fig.4d, the phase transitions from 0 to π on an energy scale that is much larger than $k_B T$ and comparable with Γ . In fact, in contrast with Yacoby et al., PRL 74, 1995, the broadening of our Coulomb line-shapes (Γ) is attributed to the lead-to-dot tunnelling, rather than to thermal broadening. It is then reasonable to expect a phase variation on the energy scale of Γ instead of $4k_B T$. Because of this, we have decided to remove the line indicating the temperature scale in Fig. 4d.

Also, we moved the discussion on the broken phase rigidity to the Supporting Information, where we expanded the text to better describe the connection between time-reversal symmetry and phase variations.

Unfortunately, we do not have data at very low magnetic field because the device was quite unstable in this regime (i.e. charge jumps occurring quite often). We speculate that the device is less prone to charge jumps at moderate magnetic fields due to the polarization of trap states at the dielectric interfaces.

3. The different phenomenology presented in D is intriguing, but also puzzling. I appreciate the fact that the authors were able to reproduce the effect by adding some complexity to their model. However, in the present

version of the paper the transition from evidences in C to those in D remains a bit obscure to me and one has the impression that fine adjustment has been introduced in the modelling. I don't want to be negative, I think the evidence is interesting and deserves publication. I would at least report the fit parameters somewhere and try and give some intuitive explanation of the mechanism leading to the change in the phase pattern.

We appreciate this remark. In this revised version of the manuscript, we report the fit parameters in the Supporting Information and we have improved the explanation of our interpretation. We support our hypothesis of multi-path transport by including a sentence in the main text on the large width of the magneto-conductance Fourier spectrum peak (with a reference to the Supporting Information). We have simplified our discussion by moving the details of the model to the Methods and conclude the section by suggesting that the exact tunnelling mechanism between the dot states and the lead could play a role in the coupling to multiple-paths. While our interpretation is speculative, but already considered in Aharony et al., Phys. Rev. B 73, 2006, it can explain why in Whiticar et al., ArXiv 1902.07085, 2019 the phase rigidity in the Coulomb valleys is not robust.

4. Datasets in C and D refer to VBG=+1.5V and -1.5V, respectively. I thus expect to have less electrons in D? Why is area less well-defined in the D configuration? Since there is less carrier density, I would expect the ring area to be MORE well-defined in configuration D than in C.

From the back-gate dependence shown in the Supporting Information, we estimated that in D the electron density is only 20% smaller than in C. We do not expect a drastic change in the definition of the area of the loop. Our observation is consistent with the scenario in which in C two consecutive dot orbitals couple to the same path, while in D four orbitals couple to different paths.

The coexistence of the two-phase behaviours is hard to explain fully. We speculate in the text that the specific gate voltages forming the tunnel barriers might affect this phenomenon, similar to the modification of the spatial parity of the electron wave function reported in Edlbauer et al., Nat. Commun. 8, 2017. In addition, as shown in Suppl. Fig. S4, the multi-path effect is observed even at a back-gate voltage of +1.5 V (but for different tunnel-gate voltages than in C).

Reviewer #3 (Remarks to the Author):

Review report - NCOMMS-19-40640

In this manuscript, the authors study the transmission phase of a quantum dot (QD) in the many-electron regime via interferometry. The experimental system consists of a network of four InSb nanowires arranged in a hashtag shape during epitaxial growth, acting as a two terminal Aharonov-Bohm (AB) interferometer, where the gate defined QD is incorporated in one of the arms. Several gates are employed to tune the carrier concentration in both the QD and the nanowire leads. The authors show that phase coherent transport is conserved through the QD in the Coulomb blockade regime as indicated by the observation of AB oscillations in the differential conductance (G) [Fig. 2]. A tight-binding model, using an expression for the total transmission from ref. [25], is used to fit the resonant peaks (dips) in G [Fig.3]. Furthermore, by tuning the gates, two regimes are identified: one with universal phase behavior (rapid phase changes of π close to the resonance follows by a phase laps in the Coulomb blocked region) [Fig.4]. The second regime, exhibit phase behavior in between the universal and the mesoscopic regime [Fig. 5]. This behavior is attributed to transport channels in the arms with different loop areas coupling to the different QD states.

This appears to be the first AB interferometry study on the InSb hashtag material system when a QD is included in one of the arms. Owing to the strong spin-orbit interaction and large g -factor, InSb is a promising material system to realise topological quantum computing. AB interferometry including a large “topological QD” has been suggested as a mean for read-out of topological qubits, but also for distinguishing between trivial and topological states in semiconductor-superconductor hybrid devices which is an important step towards topological quantum computing.

The manuscript is well-organised and easy to follow. I appreciate the clear experimental data and the close resembles between experimental and model results. Thus, overall I recommend this paper for publication in Nature communications given that the comments below are addressed:

We thank the referee for the detailed review of our work, for providing critical suggestions and the commendatory remarks.

(1) The authors point out the particular regime studied where multiple orbitals on the QD contribute to the transport has never been studied before. Can the authors comment on why this is an interesting regime to study?

The quantum-dot–interferometry experiment has been a long-studied problem. The findings of several experiments were interpreted and understood considering the relevant quantum dot energy scales (level spacing, charging energy, tunnel coupling). These quantities depend on the size of the dot. Previous experiments focused on the single-level regime (Avinun-Kalish et al., Nature 436, 2005; Yacoby et al., PRL 74, 1995; Schuster et al., Nature 385, 1997) and on the crossover between single- and few-level regime (Edlbauer et al., Nat. Commun. 8, 2017), revealing both phase lapses and phase plateaus. Our investigation in the multi-level regime – motivated by the need for mimicking large hybrid dots – completes the set of relevant cases of the dot-interferometry puzzle. We emphasised this point in a new paragraph.

Moreover, recent and near-future experiments will characterise large hybrid semiconductor-superconductor islands in nanowires (or nanowires electrostatically defined in a two-dimensional electron gas, e.g. Whiticar et al., ArXiv 1902.07085, 2019). These islands will display a rather small level spacing due to the reduced confinement. Therefore, they will most likely exhibit the universal phase behaviour due to transport via

multiple levels. Exceptional would be the case where the wire is tuned in the topological regime (see also answer to question 5 of referee #1), for which theory (L. Fu, Phys. Rev. Lett. 104, 2010; Drukier et al., Phys. Rev. B 98, 2018) predicts that the phase will exhibit a mesoscopic-like behaviour.

How would the effect of multiple lead channels masking the intrinsic transmission phase affect read out of topological qubits?

We have simplified our text in section II D and in the Discussion to clarify how this effect is detrimental for qubit read-out.

Is this effect robust when lowering the back gate further?

For reasons related to the epitaxial growth (Gazibegovic et al., Nature 584, 2017), crossing nanowire segments in hashtag-devices do not lie exactly in the same plane resulting in a non-homogeneous coupling to the back-gate. To prevent substantial density variations, we have kept the back-gate voltage around zero volt (in the range between -1.5 and +1.5 V), where the hashtag is overall populated (see Fig. S2 in the Supporting Information) and the low-temperature conductance displays quantum interference effects.

Can the authors comment on electron density in the studied regime vs the electron density needed for entering topological regime in a semiconductor-superconductor hybrid device?

We thank the reviewer for raising this point. However, discussing the density required to enter the topological regime is out of the scope of this manuscript. In order to reduce the technicalities, we have decided to avoid this topic that is thoroughly discussed in the context of hybrid devices by Winkler et al., Phys. Rev. B 99, 2018; Vaitiekenas et al., Science 367, 2020; Antipov et al., Phys. Rev. X 8, 2018; de Moor et al., New. J. Phys. 20, 2018. In general, a topological regime in InSb networks (coupled to a superconductor) can be obtained when the devices are tuned to the first few subbands. The experimental demonstration of these requirements, i.e. induced superconductivity and ballistic transport in the lowest subbands, was already provided by Plissard et al., Nat. Nanotechnol. 8, 2013 and Fadaly et al., Nano Lett. 17, 2017. In our revised version, we decided to emphasize better the link between our experiment and topological circuits without going into details (see second part of Introduction, second part of section II B, II D and in the Discussion).

(2) Fig. 2 caption: Define “G”.

We added the definition of “G” in Fig. 2.

(3) Regarding the definition of the “QD electrochemical potential”: $E_{\text{QD}} = \alpha V_{\text{PG}}$. Strictly speaking, this is only the gate contribution to the electrochemical potential. Is there an elementary charge missing or how do you define α ? Not $C_{\text{gate}}/C_{\text{total}}$?

We thank the referee for noting this error. Indeed, an elementary charge “e” is missing in the definition. It has been introduced in the revised version.

(4) Regarding Fig. 3c caption: E_c and Δ are not defined in the caption. To me it looks like the bar indicating Δ is shifted slightly too far down (I assume it is supposed to indicate the distance between the conductance line of the border of the diamond and the first parallel line). In addition, the indicated distance is 2 times the orbital spacing whereas in main text Δ is referred to as the orbital spacing. How has the

orbital spacing been extracted? Furthermore, the dashed line indicating the diamond seems to be slightly off, overestimating the E_c .

We appreciate this remark. We have adjusted the dashed lines indicating the boundaries of the diamond and the line that defines the level spacing. The length of this line is indeed two times the level spacing. We added in section II B a paragraph describing how we extracted these quantities and corrected the estimation of the charging energy. In addition, we have defined E_c and δ in the caption of Fig. 3b.

(5) Regarding estimation of electron population on the dot: For more transparency declare the estimated volume of the dot.

We added an estimate of the volume of the dot in section II B.

(6) Fig. 4: Add information on the DC bias in the caption. “magnetic flux” should be magnetic flux density of B-field. Expand the description of panel d (top) in the caption. From the caption, it is not clear why the $4k_B T$ marker is there.

We added the DC bias information and we replaced “magnetic flux” with “magnetic field”, which is the correct quantity. The caption of Fig. 4d has been expanded including information on the error bars and, following the second referee’s comments (see question 2), we have also removed the label $4k_B T$.

(7) “In order to convert between the evolving phase of $J_L = j_L \exp(i2 B \Phi)$ and the magnetic field we use the area of the loop.” Can you be more specific here and insert the full relation? Furthermore, can you include a more detailed description how you extract the transmission phase in Fig. 4d (or refer to the SI)?

We moved and re-phrased explicitly the sentence in the Methods.

In section II D, it is better explained how we extract the transmission phase (i.e. by tracking the maxima of the Aharonov-Bohm oscillations).

(8) Regarding the SI: I suggest to refer to the SI in the main text to highlight how the SI is supporting the main text?

We thank the reviewer for the input. In the revised version, we have added references to the Supporting Information where it becomes relevant (see sections II B and II D).

(9) Can you tune the QD into the few electron regime or have you a device with smaller distance between the defining gates? If so, can you see mesoscopic phase behavior or is the multiple channel effect hindering to see the mesoscopic phase behavior?

In lieu of the final qubit, we focused on a large dot. Our device could not be tuned into the few-electron regime. Indeed, this would have been of interest. However, we expect that the presence of multiple channels hinders the visibility of the mesoscopic phase behaviour.

(10) Regarding the last sentence in the conclusion: “Nevertheless, we conclude by remarking that, despite the presence of phase lapses, the transmission phase remains parity-sensitive in the vicinity of the transmission resonances. What do you mean with parity sensitive? That you see a π -shift at the resonance (change of parity)? But you cannot extract the absolute parity of the dot from the transition phase?”

We thank the referee for pointing this out. For clarity, we have decided to change the final sentence. In the new version, we stress the importance of using one-dimensional interferometers to study the transmission phase of hybrid islands. This concept is, in fact, the most relevant message of our paper.

REVIEWERS' COMMENTS:

Reviewer #1 (Remarks to the Author):

This is my second report on the manuscript "Transmission phase read-out of a large quantum dot in a nanowire interferometer" by Sebastian Heedt. After reading this second version, I am glad that the authors decided to considerably rewrite the manuscript and present a more comprehensive discussion about the nanowire interferometer, Fano physics, the multi-path regime, etc. Now, I think, this manuscript is a valuable reference stressing the difficulties towards realising more challenging experiments including parity read out of Majorana qubits in the mesoscopic regime. My overall impression is positive so I recommend this second version for publication. Before that, I only have a few minor points:

- 1) For non-experts, it can be quite difficult to actually identify a quantum dot from the image. Here, I recommend to change the colour scale (actually the "pink" cross, which is the QD, is very similar to the colour outside). They can even try to label the QD in the image. Also, a cite to the schematics in Fig.6 would be very helpful.
- 2) In page 2, first column towards the end. When discussing the cotunneling regime, Fig2c should read Fig.2d.
- 3) In section III Discussion, the sentence "In experiments, topologically trivial modes that extend over the large quantum dots and quasiparticle transport.." is probably rather obscure for non-experts. Here, the authors should include at least one extra sentence saying that zero modes without an underlying topological bulk can mimic Majoranas and hence offer parallel paths to Majorana teleportation. Here, a reference to the recent review (Prada et al, arXiv:1911.04512) discussing these zero modes seems appropriate.

Reviewer #2 (Remarks to the Author):

The authors have done a very substantial revision of their manuscript, which I believe has improved. The introduction is not aimed at a wider audience and the data analysis has been thoroughly revised. All the points that I have raised have been addressed in a satisfactory way so I am positive towards publication.

Reviewer #3 (Remarks to the Author):

To my assessment, the authors have addressed my remarks, thus I recommend publication.

Response to the reviewers' comments:

We thank all three referees for reviewing the revised version of our manuscript and for the positive feedback towards its publication. Below, we list the remarks from the referees in black and our response in blue.

Reviewer #1 (Remarks to the Author):

This is my second report on the manuscript "Transmission phase read-out of a large quantum dot in a nanowire interferometer" by Sebastian Heedt. After reading this second version, I am glad that the authors decided to considerably rewrite the manuscript and present a more comprehensive discussion about the nanowire interferometer, Fano physics, the multi-path regime, etc. Now, I think, this manuscript is a valuable reference stressing the difficulties towards realising more challenging experiments including parity read out of Majorana qubits in the mesoscopic regime. My overall impression is positive so I recommend this second version for publication. Before that, I only have a few minor points:

1) For non-experts, it can be quite difficult to actually identify a quantum dot from the image. Here, I recommend to change the colour scale (actually the "pink" cross, with is the QD, is very similar to the colour outside). They can even try to label the QD in the image. Also, a cite to the schematics in Fig.6 would be very helpful.

We slightly adjusted the colours in the image to increase the contrast between the nanowire and the QD section and added the label 'QD' in the figure. Also, we added a reference to the illustration and the schematic in the Methods section in the caption of Fig. 2.

2) In page 2, first column towards the end. When discussing the cotunneling regime, Fig2c should read Fig.2d.

We thank the reviewer for noticing this error that we have now corrected.

3) In section III Discussion, the sentence "In experiments, topologically trivial modes that extend over the large quantum dots and quasiparticle transport.." is probably rather obscure for non-experts. Here, the authors should include at least one extra sentence saying that zero modes without an underlying topological bulk can mimic Majoranas and hence offer parallel paths to Majorana teleportation. Here, a reference to the recent review (Prada et al, arXiv:1911.04512) discussing these zero modes seems appropriate.

In view of clarity, we rephrased that sentence and added the suggested reference to the review by Prada et al.

Reviewer #2 (Remarks to the Author):

The authors have done a very substantial revision of their manuscript, which I believe has improved. The introduction is not aimed at a wider audience and the data analysis has been thoroughly revised. All the points that I have raised have been addressed in a satisfactory way so I am positive towards publication.

Reviewer #3 (Remarks to the Author):

To my assessment, the authors have addressed my remarks, thus I recommend publication.

We thank all three referees very much for both reviews and the constructive feedback.